# Sparsely Connected Autoencoders: A Multi-Purpose Tool for Single Cell omics Analysis

**DOI:** 10.3390/ijms222312755

**Published:** 2021-11-25

**Authors:** Luca Alessandri, Maria Luisa Ratto, Sandro Gepiro Contaldo, Marco Beccuti, Francesca Cordero, Maddalena Arigoni, Raffaele A. Calogero

**Affiliations:** 1Department of Molecular Biotechnology and Health Sciences, University of Torino, 10126 Torino, Italy; alessandri.luca1991@gmail.com (L.A.); maria.ratto@edu.unito.it (M.L.R.); 2Department of Computer Science, University of Torino, 10149 Torino, Italy; sandro.contaldo@edu.unito.it (S.G.C.); marco.beccuti@unito.it (M.B.); francesca.cordero@unito.it (F.C.)

**Keywords:** single cell RNAseq, single cell ATACseq, sparsely connected autoencoders, gene regulatory network, transcription factor, miRNA, pseudo-bulk data

## Abstract

Background: Biological processes are based on complex networks of cells and molecules. Single cell multi-omics is a new tool aiming to provide new incites in the complex network of events controlling the functionality of the cell. Methods: Since single cell technologies provide many sample measurements, they are the ideal environment for the application of Deep Learning and Machine Learning approaches. An autoencoder is composed of an encoder and a decoder sub-model. An autoencoder is a very powerful tool in data compression and noise removal. However, the decoder model remains a black box from which is impossible to depict the contribution of the single input elements. We have recently developed a new class of autoencoders, called Sparsely Connected Autoencoders (SCA), which have the advantage of providing a controlled association among the input layer and the decoder module. This new architecture has the benefit that the decoder model is not a black box anymore and can be used to depict new biologically interesting features from single cell data. Results: Here, we show that SCA hidden layer can grab new information usually hidden in single cell data, like providing clustering on meta-features difficult, i.e. transcription factors expression, or not technically not possible, i.e. miRNA expression, to depict in single cell RNAseq data. Furthermore, SCA representation of cell clusters has the advantage of simulating a conventional bulk RNAseq, which is a data transformation allowing the identification of similarity among independent experiments. Conclusions: In our opinion, SCA represents the bioinformatics version of a universal “Swiss-knife” for the extraction of hidden knowledgeable features from single cell omics data.

## 1. Introduction

Single cell RNAseq (scRNAseq) [1,2] allows the investigation of the transcriptome from individual cells, providing an extended view of cellular differences, which can deliver a better understanding of the function of each individual cell within its microenvironment. scRNAseq has opened the route to the development of other single-cell methods, which now enable the simultaneous measurement of other data modalities like scCITEseq (Cellular Indexing of Transcriptomes and Epitopes by Sequencing), which allows the expression quantification of cell surface proteins [3]; scATACseq (Assay for Transposase-Accessible Chromatin using sequencing), which depicts accessible chromatin regions [4] and single cell spatial transcriptomics, which measures transcription expression directly on a tissue [5]. Since single cell omics analyses are becoming the most important way to investigate the functionality of healthy and disease tissues, grasping biological knowledge from single cell omics [6,7] is a mandatory subject. Since single cell technologies provide a large number of sample measurements (e.g., single cell gene expression level, single cell chromatin accessibility measurements, etc.), these methods produce the ideal input data for Deep Learning-based analyses [8,9,10]. Nowadays, autoencoders are used to moderate the excess noise level characterizing scRNAseq data [11]. Autoencoders are also used as data reduction methods for single cell transcriptomics [12]. However, an autoencoder remains a black box, from which it is impossible to depict the contribution of the single input elements.

In 2019, Gold and coworkers [13] described shallow sparsely-connected autoencoders (SSCAs) as a tool for projecting gene-level data onto gene sets. These autoencoders use a hidden single-layer with sparse connections (describing known biological relationships) in order to get a value for each gene set [13].

Recently, we have presented the sparsely connected-autoencoder (SCA) as a tool to transform single cell gene-level transcription data in metagenes based on biological features [14]. In SCA, each node represents a known biological relationship, e.g., transcription factor (TF) or miRNA. SCA receives inputs only from gene nodes associated with a biological relationship. We also showed that SCA can uncover hidden features associated with scRNAseq data being able to identify key elements for cells aggregation in single cell whole transcriptome data [14].

Here, we present an extension of SCA for scRNAseq and scATACseq. Specifically, we show that the metagene representation generated by SCA analysis provides a new environment for the functional mining of cell sub-populations depictable by scRNAseq and scATACseq analyses. Specifically, we show that SCA is a bioinformatics “Swiss-knife” for the extraction of hidden knowledgeable features from single cell omics data.

## 2. Results

To describe the peculiarities of SCA we used two scRNAseq datasets respectively made of a mixture of three (from now named: RNA-3c) and five (from now named: RNA-5c) human lung adenocarcinoma cell lines [15], and a scATACseq dataset (from now named: ATAC-5c) encompassing the same five human lung adenocarcinoma cell lines present in RNA-5c [15]. Both scRNAseq and scATACseq datasets were specifically developed as tools for benchmarking single cell data analysis methods [15].

### 2.1. Building Clusters’ Specific Pseudo-Bulk Using SCA

Single-cell expression data are zero-inflated [16], because of the prevalence of dropout events. Thus, to depict differences among clusters of a scRNAseq experiment, a straightforward solution is to create “pseudo” bulk RNA-seq data, by adding up the fragment counts of a gene across cells for each cluster, and then applying methods designed to inspect differences among samples using bulk RNAseq data, e.g., hierarchical clustering. However, such pseudo-bulk solutions reduce the distribution of gene expression across cells to a single vector. In this format, clusters’pseudo-bulk representation loses a valuable amount of information, and, due to the lack of replicates, it is impossible to perform bulk RNAseq differential expression analysis among clusters.

In our previous work [14], we have shown that SCA can be used to aggregate transcription factor (TF) target gene expression in a pseudo-value, i.e., a metagene, describing the importance of a TF in controlling its putative targets. Here, the same concept is applied to cell sub-populations depicted by clustering, Figure 1A. The overall idea of this approach is that genes characterizing a specific cells’ cluster will be the non-noise signal caught by the hidden layer of SCA (Figure 1A).

In this way, by running multiple times the SCA we can build pseudo-bulk experiments representing pseudo-replicates for each of the clusters’ gene expression. We applied this SCA analysis to the RNA-5c dataset. RNA-5c was partitioned in clusters using Seurat clustering [18], with the resolution parameter set to 0.1 (in Seurat, the resolution parameter, which ranges from 0 to 1, controls the granularity of clustering, where 1 indicates high granularity, i.e., many clusters, and 0 low granularity, few clusters). This analysis, yielded five clusters, Figure 2A.

The clusters were assigned to the corresponding bulk cell line by mean of hierarchical clustering comparing the bulk transcriptome of H2228, H1975, A549, H838 and HCC827 cell lines with the summary of the cell expression of each cluster, see Figure 2C. Subsequently, we apply the clustering results, Figure 2A, to the SCA shown in Figure 1A. The results of the pseudo-bulk analyses are summarized in Figure 1B,C. In Figure 1B, it is shown the t-Sne representation of pseudo-clusters made of 20 runs each. Pseudo-bulk clusters result to be well separated from each other. In Figure 1C, pseudo-bulk clusters well correlate with the bulk RNAseq of their corresponding cell line. Thus, this observation demonstrates the good correlation existing between SCA-pseudo-bulk clusters and the corresponding cell line bulk RNAseq transcriptome, retrieved from CCLE database [17].

### 2.2. Depicting Clusters Correspondence among Independent Experiments Using SCA Pseudo-Bulks

Single cell technology is becoming every day more used to investigate complex questions. Thus, it is becoming important to integrate the results of multiple experiments [18,20,21]. Seurat-based integration [18] is an approach frequently used to aggregate different single cells or different modalities of the same experiment. In Seurat integration, “anchors” are exploited to harmonize data from different experiments. Other methods use batch-effect correction [22] to aggregate independent single-cell experiments. Aggregation might distort single-cell data; thus, our idea is that it could be useful to simply identify clusters in common among experiments. Due to the results described in the above paragraph, we evaluated the possibility to use SCA pseudo-bulks to correlate clustering results obtained in different experiments. We compared RNA-5c and RNA-3c, which are scRNAseq experiments performed independently and constituted respectively by the following cell lines: H2228, H1975, A549, H838 and HCC827 (RNA-5c) and H2228, H1975 and HCC827 (RNA-3c). We independently built the clustering for the two datasets using Seurat clustering [18], implemented in rCASC [19]. From this analysis, we obtained five clusters for RNA-5c and four clusters for RNA-3c. RNA-3c cell line assignment, Figure 2B,D, was done and described for RNA-5c, in the previous paragraph. We independently generated SCA pseudo-bulks for the clusters of the two datasets. Pseudo-bulk counts were log_2_CPM (counts per million reads) transformed and genes expression was centered on the gene’s mean expression. The two datasets were also integrated using the Seurat integration tool [18]. The results of the analysis are shown in Figure 3.

Horizontal arrows in Figure 3A indicate the association between RNA-5c and RNA-3c. Pseudo-bulk similarity matrix also shows that RNA-3c Y1 seems to belong to HCC827 instead of H1975, as initially suggested by the hierarchical clustering in Figure 2D. Moreover, cluster Y2 shows a lighter similarity to cluster X2 and Y3 to X1. Notably, Seurat integration provides a superimposable picture to that of the SCA pseudo-bulk analysis, see Figure 3B,C. In Seurat integration, as in SCA pseudo-bulk analysis, the Y1 cluster is more near to the HCC827 cell cluster than to the H1975 cluster, which was assigned based on the hierarchical clustering in Figure 2D.

### 2.3. SCA Pseudo-Clusters as Tool in Multi-Modal Analysis

Since SCA pseudo-bulks seem to effectively depict the similarity among independent scRNAseq experiments, we tested their ability in identifying similarities in a multi-modal setting. In the GEO repository [23] are present two scATACseq experiments (GSM4224432 and GSM4224433) including 37,000 cells each, in these two experiments H2228, H1975, A549, H838 and HCC827 cell lines are mixed in a one-to-one ratio as for the scRNAseq RNA-5c experiment. The quality of the GSM4224432 experiment was very poor, thus we only used the GSM4224433 dataset for our analysis, from now on called ATAC-5c.

Unfortunately, for the ATAC-5c experiment, the association of open chromatin regions to gene loci yields only a few hundred cells with at least 400 genes loci, characterized by the presence of at least three UMI/gene loci, Figure 4A. Having such a limited number of cells, harboring gene-associated chromatin conformation information does not guarantee that all the five cell lines are equally presented in the ATAC-5c experiment. The clustering of these cells depicts six clusters, see Figure 4B. Those clusters are used to generate SCA pseudo clusters, and the similarity of such clusters with respect to RNA-5c is calculated, see Figure 4C. Although the similarity among RNA-5c and ATAC-5c data is quite blurred; in Figure 4C, we can still depict a weak similarity among a1/X1, a4/X3 and a6/X5 clusters.

### 2.4. SCA as Tool for the Detection of Regulatory Gene Hubs

In our previous paper on SCA [14], we used the output of SCA based on TFs or miRNA targets to reconstruct at least some of the clusters generated using the full gene matrix. An SCA based on TFs or miRNA targets generates a hidden layer made of metagene pseudo-expression. Each metagene is a value derived from the level of expression of the genes, whose expressions are controlled by a specific TF or miRNA. Thus, in the case an SCA pseudo-expression matrix can reconstruct a cluster, previously generated using all genes’ expression values, this is a strong indication of how the meta-features, i.e., TFs or miRNA, are important players in the formation of that specific cluster. Here, we show that we can grasp more information clustering the output of an SCA based on TFs or miRNA targets. The peculiarity of SCA is that its hidden layer encodes the most informative parts of a dataset. Thus, running multiple times an SCA and performing the cumulative sum of the output results there is a progressive increase of important feature signals as instead the not informative portion of the signals remain near a background level. Using such an approach we clustered the output of TFs and miRNA SCA, for RNA-5c, Figure 5A,B.

In Figure 5A, we can note that clustering based on SCA meta-genes provided more compact clusters with respect to the analysis done using the full gene set, Figure 2A.

Furthermore, metagene representation has the advantage of simplifying the understanding of the main functional characteristics of each cluster, since it brings the view of the data to a domain of knowledge-rich information, e.g., transcription factors or miRNAs. To further support this point, we extended the characterization of the clusters shown in Figure 5A, focusing our analysis on the 100 most robust cluster-specific TF markers, depicted by COMET analysis [24]. COMET is a general non-parametric statistical framework for the identification of candidate marker panels consisting of one or more genes for cell populations of interest identified with single-cell RNAseq data. Within the 100 most effective cluster-specific TF markers, depicted in each cluster, we selected those TFs characterized by a log_2_ fold change greater than 1 (N.B. log_2_ fold change indicates the expression difference for a cluster-specific TF marker with respect to its expression in the other clusters). Then, we challenge these TFs for their ability to generate regulative networks, using OmicsNet [25] framework. OmicsNet is a web-based tool allowing the creation of different types of molecular interaction networks and providing a visual exploration in a three-dimensional (3D) space. Indeed, a subset of TFs selected from each of the five clusters, Figure 5A, can generate gene networks of various sizes, Appendix A. We further inspected the networks using Gene Ontology (GO) enrichment. The up-modulated TFs networks associated with clusters 1 and 4, Appendix A, are statistically significantly enriched in the “cell proliferation” GO term. On the other hand, cluster 2 and 5 down-modulated networks, Appendix A, are also enriched in the “cell proliferation” GO term. Thus, suggesting that cells in clusters 1 and 4 proliferate more successfully than cells in clusters 2 and 5. Notably, cells in cluster 3 have as top-ranked GO term the response to chemical stimuli, suggesting that these cells might be particularly sensitive to changes in the tumor microenvironment.

In Section 2.2, using SCA pseudo-bulk we showed that clusters 4 and 5 from the RNA-5c experiment share some level of similarity, Figure 3, although the two clusters belong to different cell lines. To evaluate if miRNA representation of the RNA-5c experiment could add some extra information to the similarity observed for clusters 4 and 5, Cluster 4, in Figure 5B, is inspected for the presence of intersection among the top 100 most robust cluster-specific miRNAs, detected by COMET analysis in all the other clusters. Notably, only an overlap of four miRNAs (miR-10b-5p, miR-125b-5p, let-7g-5p, miR-24-3p) between cluster 4 and cluster 5 is detectable. Using OmicsNet analysis [25], we observed that all four miRNAs were characterized by having CDKN2A as a common target. Notably, the average log_2_ fold change associated with miR-10b-5p, miR-125b-5p, let-7g-5p, miR-24-3p in cluster 4 is four times higher than those in cluster 5. CDKN2A is a well-known tumor suppressor gene [26], this observation predicts a stronger proliferation activity associated with the cells in cluster 4 with respect to cluster 5, due to the higher expression of the miRNAs negatively controlling CDKN2A expression in cluster 4 cells. Such an observation is in frame with the different proliferation behavior predicted by the OmicsNet analysis of TF markers, Appendix A.

We also ran the TF and miRNA SCA analysis on RNA-3c. In this case, we obtained three homogeneous clusters. The clusters Y1, Figure 2D and Figure 3D, which show a discrepancy in cell type assignment, were embedded in clusters T1 and m1, Figure 6A,B. Thus, confirming that cluster 1, Figure 2B, belongs to the HCC827 cell line.

## 3. Discussion

An autoencoder is composed of an encoder and a decoder sub-model. The encoder condenses the input, and the decoder tries to reconstruct the input from the condensed version provided by the encoder. Usually, after training, the encoder model is saved, and the decoder is discarded. The idea of using non-fully connected autoencoders is very recent. In 2019, Gold and coworkers [13] proposed the Shallow Sparsely-Connected Autoencoder (SSCA) and the Variational Autoencoder (SSCVA), as promising tools, which can be exploited in the identification of transcription factors with differential activity among conditions or cell types. In SSCA/SSCVA the decoder is not fully connected, but it is made of meta-features, showing specific relation with respect to the input genes. More recently, we [14] showed that a type of autoencoder similar to those described by Gold and coworkers, the Sparsely Connected Autoencoder (SCA), can be used to reconstruct at least some of the clusters, previously depicted by clustering analysis of the full genes set of single cell RNAseq data. The peculiarity of SCA [14] is that the decoder model is not discarded, but it is used to grab more functional knowledge from the input dataset. This is possible because, due to the specific relations existing among input genes and decoder meta-features, the decoder is not a black box anymore. Thus, SCA [14] can provide new incites in the biological meaning of the reconstructed single cell RNAseq clusters.

In this paper, we extend the mining strength of SCA. Specifically, we show in Figure 1 that SCA can be used to reconstruct multiple pseudo-replicates representing the average expression of scRNAseq clusters. These cluster-based pseudo-replicates are a modeled average representation of the expression of the cell genes present in each cluster, previously defined by an scRNAseq analysis on the full gene set. SCA pseudo-bulks have the advantage of not being zero-inflated [27], making them ideal to depict differentially expressed genes among different scRNAseq clusters using conventional RNAseq tools [28,29]. Furthermore, we observed that SCA clusters’ pseudo-bulks work effectively in depicting the relationship among scRNAseq data coming from different experiments. Specifically, we showed, in Figure 3, that SCA clusters’ pseudo-bulks have the same integrative power as Seurat [30]. However, the use of SCA pseudo-bulks in multi-modal data analysis, seen in Figure 4, requires further efforts to fully understand its integrative power, due to the limited quality of the scATACseq dataset (ATAC-5c), which was used in the comparison with scRNAseq (RNA-5c). Specifically, we need to run clusters’ pseudo-bulk comparisons on a dataset in which the two modalities, i.e., scRNAseq and scATACseq, are collected from the same cell, to estimate how strong is the correlation between transcriptomics and open chromatin. We have recently found on GEO the dataset GSE151302 [31], which is made by five single nucleus ATAC (snATAC-seq) and RNA (snRNA-seq) sequencing and provides cell-type-specific chromatin accessibility and transcriptional profiles of adult human kidneys. We are in the process to analyze these data.

We have also extended the functional mining power of the SCA developed in [14]. In the present implementation, the decoder layer is based on transcription factors (TFs) or miRNAs, i.e., each node of the hidden layer represents a TF or a miRNA. Each node of the decoder layer is connected only to the corresponding input target genes. SCA is executed multiple times and the resulting decoder layers are aggregated by a cumulative sum. Because the hidden layer of an autoencoder can provide a representation of the input data in which noise is at least partially discarded, summing multiple runs of the hidden layer allows highlighting genes playing an important role in cell subpopulations with respect to genes that are non-specifically modulated. We observed that the clustering of TFs meta-features, Figure 5A, provides more homogeneous cell subpopulations than those depictable using the raw data, Figure 2A. Furthermore, the artifactual identification of four clusters for three cell lines, in Figure 2B, is completely solved in Figure 6, where both the number of clusters and the cell lines are perfectly matching. Furthermore, the results generated using SCA TF meta-genes, in Figure 5A, perfectly agree with the observation resulting from the integration with pseudo-clusters, Figure 3A, and with Seurat, Figure 3B,C. SCA TF meta-genes clustering, SCA-pseudo-bulks and Seurat integration highlight the association of cluster Y1, Figure 2B, with cell line HCC827 instead of H1975, observed from the clustering of raw data, Figure 2D. Furthermore, the analysis of the regulative networks depicted by the analysis of SCA TFs metagenes, Appendix A, and the overlap detected, within the SCA miRNAs analysis, between cluster 4 and 5 of the RNA-5c experiment highlights that SCA metagene representation seems to be an interesting new environment to investigate the functional characteristics of scRNAseq clusters. Thus, SCA meta-features provide a more robust representation of the cell subpopulations depictable by clustering of single cell data.

Taken together our data highlight some interesting and useful features of SCA as data mining tools.

## 4. Materials and Methods

### 4.1. scRNAseq Benchmark Preprocessing

Preprocessing was performed with the rCASC package version 5.2.4 [19]. Count matrix for scRNAseq, produced by a mixture of three (H2228, H1975 and HCC827) and five human lung adenocarcinoma cell lines [11] (H2228, H1975, A549, H838 and HCC827) were respectively retrieved from GSM3022245 and GSM3618014 datasets available at GEO database [22]. Counts were adjusted by SAVER version 1.1.2 [32], to provide accurate expression estimates for all genes. Subsequently, count tables were filtered to remove low quality cells, i.e., those cells with less than 250 genes called *“present”* (a gene is called present if supported by at least three UMIs, we use the three UMIs/gene as a threshold because it allows to moderate the effect of sequencing errors in UMI counting when up to two UMIs are used to call present a gene) for the five cell line dataset, and those cells with less than 100 genes called present for the three cell line dataset.

Then, for the five cell lines dataset, we retained only the 2500 most variant genes out of the 5000 most expressed. The resulting dataset made of 3904 cells and 2500 genes will be called from now RNA-5c. The three cell lines dataset were filtered to retain the same 2500 genes selected in RNA-5c, from now this dataset is called RNA-3c.

### 4.2. scRNAseq Cell Type Association

TPM expression data for H2228, H1975, A549, H838 and HCC827 cell lines were retrieved from the CCLE repository (https://depmap.org/portal/download/, accessed on 10 October 2020). Bulk expression cell lines were row-mean centered.

RNA-5c and RNA-3c were independently clustered using Seurat, implemented in rCASC. The analysis was done using the Seurat resolution parameter set at 0.1 to generate a number of clusters as much like the expected cell line number. From Seurat clustering, we obtained five clusters for RNA-5c and four clusters for RNA-3c. Subsequently, each cluster was converted in pseudo-bulk CPM expression and the CPM data were row-mean centered.

Mean centered bulk cell lines and cluster pseudo-bulks were combined and clustered using Morpheus (https://software.broadinstitute.org/morpheus/, accessed on 5 September 2021), hierarchical clustering was performed using Euclidean distance and average linkage, see Figure 2.

Integration of RNA-5c and RNA-3c was performed using Seurat integration implemented in rCASC (Seurat resolution = 0.1).

### 4.3. scATACseq Benchmark Preprocessing

The scATACseq processed data for the same mixture of five cell lines used in RNA-5c were retrieved from the GEO repository: GSE142285. GSE142285 includes two samples GSM4224432 and GSM4224433. Open chromatin regions for samples GSM4224432 and GSM4224433 were associated with human genes, defined as the gene locus in ENSEMBL hg38 version 100 GTF file. Subsequently, cells characterized with at least 400 genes, i.e., a gene is called present if supported by at least three UMIs. Respectively 98 and 597 cells out of 37,000 passed the above filter in GSM4224432 and GSM4224433. The 597 cells from the GSM4224433, now named ATAC-5c, were subsequently analyzed by Seurat clustering implemented in rCASC.

### 4.4. Software Implementation

All software described in the present paper was implemented in the rCASC package version 5.2.4 [19].

### 4.5. SCA-Pseudo-Clusters Generation

SCA pseudo-clusters, Figure 1 and Figure 3 were generated using the rCASC function *autoencoder4pseudoBulk* (parameters: permutation = 20, nEpochs = 1000).

### 4.6. Seurat Clustering

Seurat clustering, Figure 2A,B, were done using the Seurat clustering function embedded in rCASC: *seuratBootstrap* (parameters: resolution = 0.1, nPerm = 90, PCADIM = 10, seed = 111). The optimal number of PCA components to be used in the clustering, i.e., PCADIM parameter, was depicted using the rCASC function *seuratPCAEval*.

Seurat clustering, Figure 4B, was done using the Seurat clustering function embedded in rCASC: *seuratBootstrap* (parameters: resolution = 0.6, nPerm = 90, PCADIM = 10, seed = 111).

### 4.7. Hierarchical Clustering

Hierarchical clustering, Figure 2C,D was performed using the online Morpheus tool from Broad Institute (https://software.broadinstitute.org/morpheus/, accessed on 5 September 2021). Figure 3A and Figure 4C Pearson similarity were performed using the Morpheus tool.

### 4.8. Seurat Integration

Seurat integration, Figure 3B,C, was performed using the Seurat embedded in rCASC, using the *seuratIntegration* function (parameters: seed = 111, k = 0.1).

### 4.9. SCA-Metagenes

SCA TF and miRNA metagenes, Figure 4 and Figure 5, were build using the function *autoencoder4clustering* (parameters: permutation = 100, nEpochs = 1000). Clustering on the matrix resulting from the sum of the dense spaces was done with *seuratBootstrap* (parameters: resolution = 0.1, nPerm = 40, PCADIM = 5, seed = 111).

## 5. Conclusions

Sparsely Connected Autoencoders (SCA) represent a new way of looking at Deep Learning tools. Specifically, SCAs offer the opportunity to transform data in a controlled way to grasp from single cell data hidden biological information like relations among cell subpopulations and simplify the inspection of functional relationships among regulatory elements as transcription factors or miRNAs. Furthermore, the peculiar ability of the autoencoder to retain only the important part of a signal can help in discriminating between true differences among cell subpopulations and clustering overfitting.

## Figures and Tables

**Figure 1 ijms-22-12755-f001:**
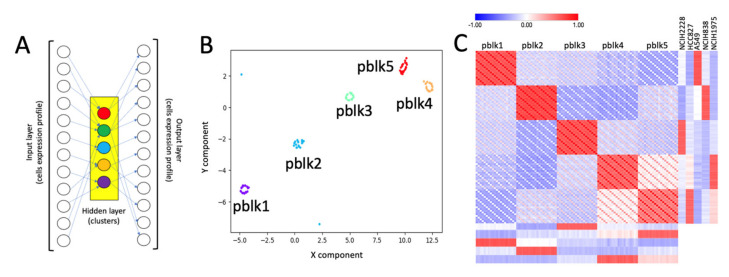
Pseudo-bulk clusters generated by SCA. (**A**) Structure of the SCA used to generate pseudo-bulk RNAseq data from each sub-population (cluster), depicted by clustering of RNA-5c dataset, Figure 2A. The pseudo-bulks are generated using the hidden layer data, repeating multiple times the SCA runs. (**B**) t-Sne output of 20 runs of SCA. It is notable that multiple runs of SCA produce pseudo-bulks, which are clustering around the centroid of each cells’ sub-population, i.e., the circular/oval structures observable in the tSne plot for the 20 runs of SCA. (**C**) Row-mean centered CPM (counts per million of reads) expression for pseudo-bulks (**B**) is combined with row-mean centered TPM (transcripts per million of reads) expression of the bulk RNAseq for H2228, H1975, A549, H838 and HCC827 cell lines, retrieved from the CCLE repository [17]. The plot shows the Pearson correlation matrix of the two datasets. It is notable that pseudo bulk expression correlates with the RNAseq bulk experiment for the five cell lines present in the RNA-5c experiment.

**Figure 2 ijms-22-12755-f002:**
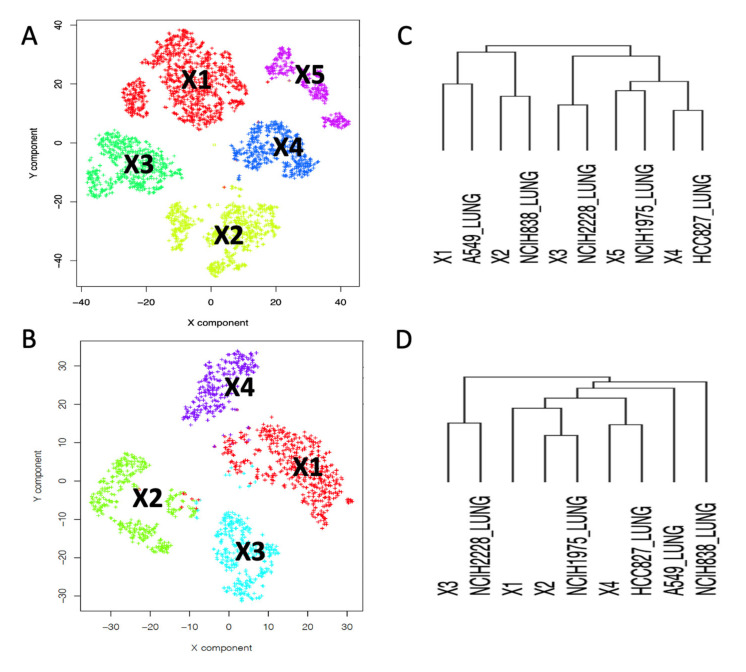
Assignment of cell line type to clusters generated with Seurat [18], implemented in rCASC [19]. (**A**) Five clusters were generated with Seurat (resolution = 0.1), using 2500 genes selected as the most variant within the 5000 most expressed in the RNA-5c experiment. (**B**) Four clusters were generated with Seurat (resolution = 0.1) using RNA-3c experiment, selecting the same 2500 genes used in (**A**). (**C**) RNA-5c hierarchical clustering (Euclidean distance, average linkage) of log_2_CPM clusters’ pseudo-bulk expression, row-mean centered, combined with the log_2_TPM, row-mean centered, for CCLE lung cell lines A449, NCIH838, NCIH2228, NCIH1975 and HCC827. (**D**) RNA-3c hierarchical clustering same parameters used in (**C**). In the RNA-5c experiment, hierarchical clustering of the clusters’ pseudo-bulks, combined with the expression data retrieved for the bulk RNAseq for the five cell lines (**C**) well correlate to each other. Only in the RNA-3c experiment, cluster X1 (**D**) shares similarity with more than one cell line (NCIH1975 and HCC827).

**Figure 3 ijms-22-12755-f003:**
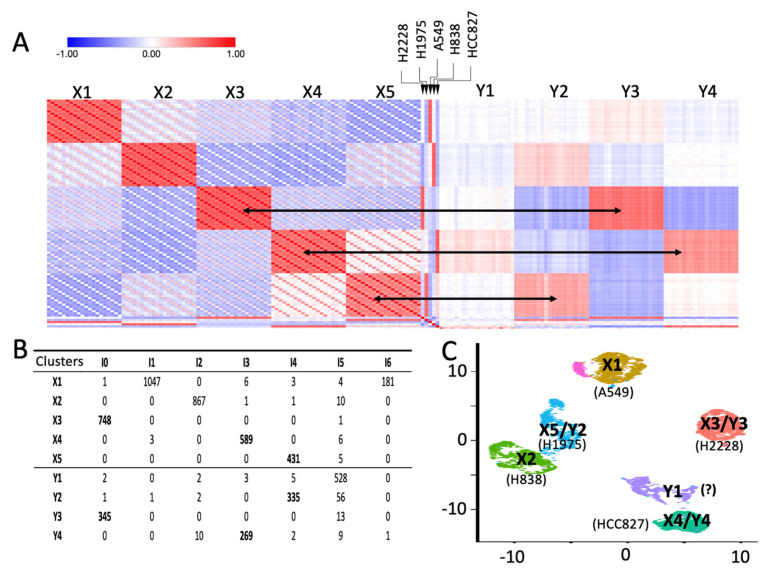
Comparing independent datasets using SCA pseudo-bulk. (**A**) Pearson similarity matrix generated comparing RNA-5c (X1–5) and RNA-3c (Y1–4) clusters, together with the bulk cell lines transcriptome. Black arrows associate clusters with the higher similarity depicted among clusters. It is notable that SCA pseudo-bulk experiments are very effective in capturing the similarity among clusters from independent experiments. (**B**) Seurat integration table. On the columns are shown the integration clusters and on the rows are the number of cells from each RNA-5c and RNA-3c cluster present in the integration clusters. (**C**) UMAP (Uniform Manifold Approximation and Projection) plot of the Seurat integrated clusters. Y1 cell line association is indicated with a question mark since, in Figure 2D, Y1 seems to be associated with H1975, as instead, by Seurat integration and SCA, pseudo-bulk Y1 shows higher similarity for HCC827 than to H1975.

**Figure 4 ijms-22-12755-f004:**
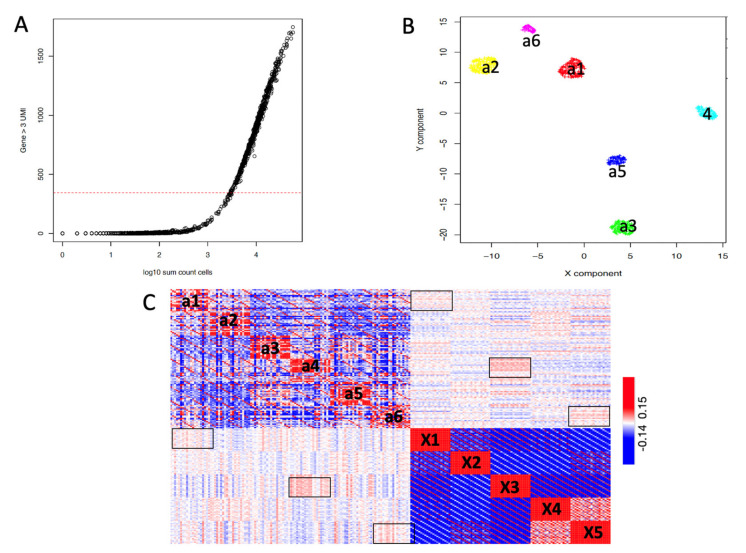
ATAC-5c comparison with respect to RNA-5c. In the ATAC-5c experiment, ATACseq regions are associated with genes (counts associated with each gene indicate the number of reads detected by ATACseq within the genomic coordinates of each gene locus). (**A**) Cells are represented based on the number of genes supported by at least three UMI and plotted with respect to the total number of UMI in each cell. It is notable that the ATAC-5c experiment has a lot of cells where the number of open chromatin regions associated with genes is very little. In this analysis, we only used cells supported by at least 400 open chromatin regions associated with coding genes. (**B**) Seurat clustering (resolution: 0.6) of the cells’ ATACseq counts identified six clusters. (**C**) SCA pseudo-bulk Pearson similarity among ATACseq clusters (clusters a1–6) and scRNAseq (clusters X1–5, Figure 2A). Square boxes indicate the highest similarity among clusters: a1/X1, a4/X3 and a6/X5. The overall similarity among clusters of the two modalities is quite low, but still, some level of similarity can be depicted (square boxers).

**Figure 5 ijms-22-12755-f005:**
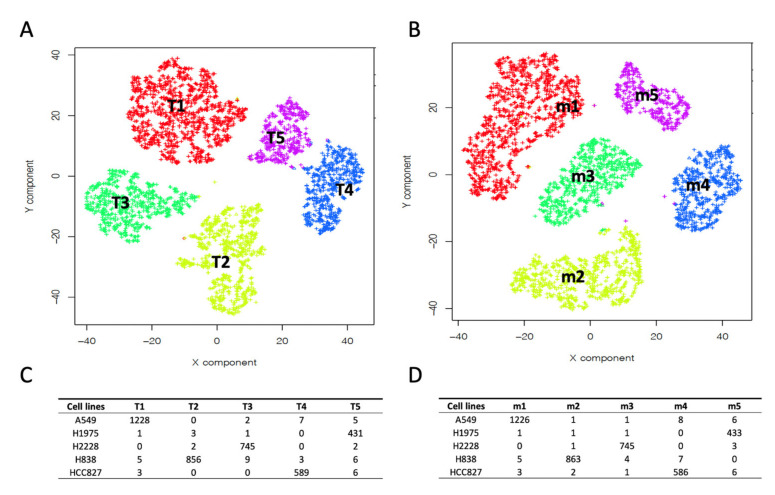
Clustering of cells in RNA-5c using SCA meta-genes. (**A**) Results of the clustering of SCA meta-genes based on TFs, (**B**) Results of the clustering of SCA meta-genes based on miRNA, (**C**) Summary of the cells, in the various clusters based on TFs, associated with the cell lines present in RNA-5c experiment. (**D**) Summary of the cells, in the various clusters based on miRNAs, associated with the cell lines present in the RNA-5c experiment.

**Figure 6 ijms-22-12755-f006:**
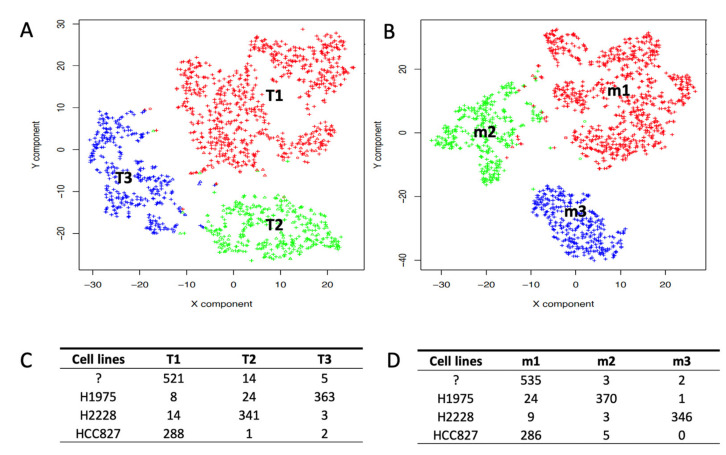
RNA-3c, clustering of cells using the SCA meta-genes. (**A**) SCA using TFs targets, (**B**) SCA using miRNA targets, (**C**) clusters to cell association for A, (**D**) clusters to cell association for B.

## Data Availability

All data used for the generation of the figures shown in this paper are available at figshare.com: https://figshare.com/projects/Sparsely_Connected_Autoencoders_a_multi-purpose_tool_for_single_cell_OMICs_analysis/123226, accessed on 10 November 2021.

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
