# Peer review of "Sparsely Connected Autoencoders: A Multi-Purpose Tool for Single Cell omics Analysis"

_ijms, 2021, doi:10.3390/ijms222312755_

Round 1

Reviewer 1 Report

The paper of Luca Alessandri and co-authors titled "Sparsely Connected Autoencoders: a multi-purpose tool for 3 single cell OMICs analysis" seems to be dedicated to the single cell OMICs analysis.

However, for me it was quite unclear, what genre of scientific publication this paper belongs to. Is it a novel method approbation? Is it a novel software description? Is it a software applying to get novel biological results? The presented manuscript lacks important things whatever the paper genre was chosen by the authors.

I would like to see more evidence for statements like this: "Thus, SCA [10] can provide new 235 incites in the biological meaning of the reconstructed single cell RNAseq clusters".  What "biological meaning" kept the authors in mind? There are too many general words in the text, which can't be understood without knowing the context of the narrow field. Moreover, figures presenting clusters require better explanation.

There are also many minor issues like undefined abbreviations, uncorrect English phrases, incorrect use of the articles etc.

Author Response

Reviewer 1

Q1: The paper of Luca Alessandri and co-authors titled "Sparsely Connected Autoencoders: a multi-purpose tool for 3 single cell OMICs analysis" seems to be dedicated to the single cell OMICs analysis.

However, for me it was quite unclear, what genre of scientific publication this paper belongs to. Is it a novel method approbation? Is it a novel software description? Is it a software applying to get novel biological results? The presented manuscript lacks important things whatever the paper genre was chosen by the authors.

A1: The paper refers to an extension of the mining features of the sparsely-connected-autoencoders (SCA), previously described by us. Such extension improves the ability of the software in providing new functional information on single cell clusters depicted by scRNAseq technology. To fulfil the indication of the reviewer we extended the introduction, the results and the discussion sections, please see all red parts inserted in the manuscript.

Q2: I would like to see more evidence for statements like this: "Thus, SCA [10] can provide new incites in the biological meaning of the reconstructed single cell RNAseq clusters". 

A2: We have extended the analysis of cluster-specific TF markers for the SCA clustering shown in Figure 5A, please see the red part at the end of paragraph 2.4

Q3: What "biological meaning" kept the authors in mind?

A3: We refer to the ability of the SCA metagenes representation to simplify the understanding of the main functional characteristics of each cluster, since such metagene representation brings the view of the data to a domain of knowledge more reach of information, e.g. transcription factors or miRNAs. Please see red section at the end of paragraph 2.4.

Q4: There are too many general words in the text, which can't be understood without knowing the context of the narrow field. Moreover, figures presenting clusters require better explanation.

A4: We revised the overall article to clarify concepts and we provided a more extended explanation for the figures.

Q5: There are also many minor issues like undefined abbreviations, uncorrect English phrases, incorrect use of the articles etc.

A5: We revised the overall article, fixing undefined abbreviations and revising English.

Reviewer 2 Report

The text is not well prepared. Terminology is mixed in parts (like OMICs or omics). I think the authors extensively cite their previous publications (3 paper Alessandri et al., 2012, 2012 and 2021). Need undeline novelty of the presented work, extend discusssion. I'd recommend add more recent references on scRNA-seq. This manuscript needs rather major than minor revision.

Author Response

Reviewer 2

Q1: The text is not well prepared. Terminology is mixed in parts (like OMICs or omics).

A1: We revised the overall article to provide homogeneous terminology and fixing undefined abbreviations.

Q2: I think the authors extensively cite their previous publications (3 paper Alessandri et al., 2012, 2012 and 2021). Need underline novelty of the presented work, extend discussion.

A2: Please note that sparsely connected autoencoders are a very new type of autoencoders and only two papers are referring to them our Alessandri et al 2021 and Gold 2019. We extended results and discussion. Please see answer A1 to reviewer 1.

Q3: I'd recommend add more recent references on scRNA-seq. This manuscript needs rather major than minor revision.

A3: We revised the references on scRNAseq.  We also added more references for the other single cell technologies described in the introduction. Furthermore, we revised the introduction to provide a better background to single cell technologies.

Round 2

Reviewer 1 Report

The authors have addressed my comments and suggestions.

Reviewer 2 Report

Dear Authors,

I'm sorry for technical mistake from my side as the reviewer. I put only short comments to this window 'Comments and Suggestions for Authors'. I have long list of comments on the text - not answered yet. But the manuscript has been revised significantly and half of my remarks is not actual for current version. However some phrase need be rewritten such as 'For research articles with several authors, a short paragraph specifying their individual contributions must be provided.' - It is clearly a draft text from a template.

Please check the remarks below (for previous version) and ensure that all the phrases are updated:

---

Terminology is mixed and sometimes it is not standard. We may keep ‘RNAseq’ instead of ‘RNA-seq’

But need write’omics’ not ‘OMICs’

The abbreviations should be in full from the beginning.

I think the authors extensively cite their previous publications (3 paper Alessandri et al., 2012, 2012 and 2021). Need underline novelty of the presented work, extend discussion, add recent references to the reference list.

I think ‘omics’ should be in lowercase starting from the title and the abstract.

Abstract should have no abbreviations.

Phrase ‘Single cell multi-OMICs is a new tool aiming to provide new  incites in the complex network of 

events controlling the functionality of the cell.;’ should be updated.

It is not clear what is the problem - bioinformatics method or just technology in general, or program name.

Typo ‘.;’

The terms ‘Deep Learning’ and ‘Machine Learning’ could be in capital.

‘AE’ abbreviation is redundant in the abstract.

‘AE are very powerful in data compression and noise removal.’ - this sentence should changes, remove ‘AE’ and put it before the sentence discussion a decoder.

Decoding information is common problem for any predictor.

In the Abstract ‘We have recently developed a new class of autoencoder’ - need indicate application of this function, if not cite in the Abstract.

Change phrase ‘or impossible, miRNA expression...’ why it is impossible?

Keywords - ‘pseudo-bulk’ - what is it?

Add word ‘data’ or ‘clustering’ to ‘pseudo-bulk’ in the keywords list.

Line 35: ‘CITEseq’ - need give it in full

Line 37: ‘OMICs’ -> omics - here and through the text

Line 40: [6-8] - bulk citation, extend the phrase

Line 41 ‘recently us  [10]’ - please rephrase

Lines 46- 49: ‘In our recent paper  [10]’ ... ‘Alessandri et al. paper [10]’ -

Multiple citation of [10]. Rephrase, make it shorter. One citation is enough.

Line 51: ‘bioinformatics “Swiss Army knife” ‘ -

I think this term should be explained. “Swiss-knife” is already a term, not need word ‘Army’. I’d recommend add citation. Explain the concept.

Line 55: ‘three (RNA-3c) and five (RNA-5c)’ - this phrase should be changed.

Write like we name this dataset as ‘RNA-3c’, at least. It is new term, need avoid it. Need give names of the cell lines. IT is not clear from the beginning that 5 cell lines just have two new cell lines.

Line 57: Typo ‘,.’

Lines 63 and 67: ‘“pseudo” bulk’ and ‘pseudo-bulk’ - need give exact phrase, that it is authors’ definition

The legend for Figure 2 should be updated.

It is not clear what are ‘Pseudo-bulks generated by SCA’ - add word ‘clusters’. Give note for ‘pblk1’, ‘pblk2’ etc. on the figure

Give abbreviations for t-Sne and TPM (lines 79-81).

Legend for Figure 2 - avoid abbreviations ‘RNA-5c’ and ‘RNA-3c’

For panel (B): ‘2500 genes selected for RNA-5c’ but ‘RNA-3c clustering’ - it is a mixture, or what are the 2500 genes?

Line 109: ‘Thus, demonstrating..’ - the phrase is not complete in English

Line 125: ‘H2228, H1975, A549, H838 and HCC827  (RNA-5c)  and H2228, H1975 and HCC827  (RNA-3c).’ -

Need rephrase, comment about the cell lines, how they are related to each other, what mean new added cell lines A549, H838?

Line 27: ‘Gold indicated’ - add ‘Gold et al’ or ‘Gold and co-authors’

The Methods section came late in the text. Still some parameters selection is not clear. All the details are referred to previous authors’ publication.

The filtering of 2500 genes only looks like artificial procedure. Such gene selection may disturb the program testing result.

Conclusion is rather short. Need support it by references, by own results.

Lines 351-352 ‘For research articles with several authors, a short paragraph specifying their  individual contributions must be provided. The following statements should be used’  -

The text is not formatted!

Line 261: ‘Declare conflicts of interest or state...’ - it is so bad formatting - no respect to the journal if the authors even can’t prepare and format text.

Minor - ‘RNAseq’ - I believe it should be written as ‘RNA-seq’, but may keep as is

------------ end of remarks for old version ---

Please update the Abstract text and check the terms used in the text.

“Swiss Army knife” - could be commented as a ‘universal “Swiss-knife” tool’ or like that.

Term ‘OMICs’ should be commented as ‘OMICs (genomics, transcriptomics)’ to follow the standards in the current literature.

Add volume and page numbers for references 4 and 21.

Overall, I recommend add references for recent literature by 2021.

The text need editing and additional revision (my estimate is between major and minor revision).